# Rising prevalence of depression and widening sociodemographic disparities in depressive symptoms among Filipino youth: findings from two large nationwide cross-sectional surveys*

Joseph H. Puyat[1,2] , Divine L. Salvador[3], Anna C. Tuazon[3] and Sanny D. Afable[4]

[1]Centre for Advancing Health Outcomes, Providence Health Care, Vancouver, BC, Canada; [2]School of Population and Public Health, Faculty of Medicine, University of British Columbia, Vancouver, BC, Canada; [3]Department of Psychology, College of Social Sciences & Philosophy, University of the Philippines Diliman, Quezon City, Philippines and [4]School of Geography and Sustainable Development, University of St Andrews, Scotland, UK

## Research Article

**Keywords:**
child mental health; collective mental health; COVID; depression; developing countries

**Corresponding author:**
J. H. Puyat;
Email: jpuyat@advancinghealth.ubc.ca

*This article has been updated since original publication. A notice detailing the change has also been published

## Abstract

Youth depression is a critical target for early intervention due to its strong links with adult depression and long-term functional impairment. In low- and middle-income countries (LMICs) like the Philippines, limited epidemiological data hampers mental health service planning for youth. This study analyzed nationally representative survey data from 2013 (n = 19,178) and 2021 (n = 10,949) to estimate the prevalence of moderate to severe depressive symptoms (MSDS) among Filipinos aged 15–24 years, using the 11-item version of the Center for Epidemiologic Studies Depression Scale. Survey-weighted analyses revealed that MSDS prevalence more than doubled from 9.6% in 2013 to 20.9% in 2021. The rise was most pronounced among females (10.8% to 24.3%), non-cisgender or homonormative individuals (9.7% to 32.3%), youth with primary education or less (10.8% to 26.5%), youth from economically disadvantaged households (10.6% to 25.1%) and youth who were separated, widowed or divorced (18.3% to 41.3%). Disparities in MSDS also widened over time, with some groups bearing a disproportionate burden. These findings underscore the need to expand accessible, high-quality mental health services for youth in LMICs, such as the Philippines. Continued monitoring and targeted interventions are essential to address the rising burden of depression, particularly among underserved and disproportionately affected groups.

## Impact statement

The rising prevalence of moderate to severe depressive symptoms (MSDS) among Filipino youth is a pressing public health concern that demands immediate attention. This study highlights a troubling trend, with MSDS rising from 9.6% in 2013 to 20.9% in 2021. This increase not only underscores the deteriorating mental health landscape for young people in the Philippines but also signals a potential crisis that could have long-term implications for individuals and society as a whole.

Particularly alarming is the disproportionate impact on specific sociodemographic groups, including females, non-cisgender individuals, and those from economically disadvantaged backgrounds. These disparities indicate that certain populations are more likely to experience depression and face greater barriers to accessing necessary mental health services. The findings call for targeted interventions that address these inequities and provide tailored support to those most at risk.

In low- and middle-income countries settings like the Philippines, where mental health services are often limited, this research serves as a critical foundation for policymakers and health professionals. It highlights the urgent need for expanding accessible, high-quality mental health services to cater to the unique needs of youth. It also underscores the necessity for ongoing research to monitor trends in MSDS and evaluate the effectiveness of interventions aimed at underserved populations. Addressing youth depression not only improves individual well-being but also promotes broader societal health, economic productivity and social stability.

## Introduction

Youth depression is a major concern and a critical focus for early intervention because it tends to foreshadow chronic, recurrent mental health problems in adulthood (Weersing et al., 2017). In the Philippines, the limited data collected before and during the COVID-19 pandemic, indicate that a significant number of Filipino youth are affected by depression (Alayon, 2021;

Alejandria et al., 2023) and other mental health problems, such as suicide (Alayon, 2021; Garcia, 2019; Puyat et al., 2021).

Using data from a nationwide cross-sectional survey of Filipino youth in 2013, Puyat et al. (2021) estimated the prevalence of moderate to severe depressive symptoms (MSDS) to be 8.9% for young Filipino adults, with the prevalence being higher for females (10.2%) than males (7.6%). The study also found that the prevalence of MSDS differed by key demographics such as educational attainment, civil status and living in urban *versus* rural areas. These findings underscore how structural and social factors such as poverty, limited education and marginalization, give rise to and perpetuate mental health disparities.

The social determinants of health (SDH) framework helps explain these disparities by emphasizing that health outcomes, including mental health, are shaped by broader social, economic and environmental conditions in which people are born, live, work and age (Allen et al., 2014). Within this framework, depression is not merely an individual concern but a reflection of systemic inequities that require structural solutions. Effective mental health strategies must, therefore, extend beyond individual-level interventions to address the underlying social conditions that contribute to psychological distress, such as economic insecurity, inadequate access to education and social exclusion.

The COVID-19 pandemic further exposed and deepened existing social and economic inequalities, amplifying the very conditions that contribute to mental health disparities. In addition to exacerbating poverty and marginalization, the pandemic brought about widespread social and physical restrictions that disrupted work, daily life and recreation. These disruptions not only restricted access to health services, but also weakened individuals' and communities' ability to cope and thrive (WHO, 2021). In the Philippines, government-imposed lockdowns and restrictions led to closure of local government facilities that offered counseling services, severely reducing access to mental health services. School closures further compounded these issues by limiting access to school-based mental health services, including school counselors, teachers and peer support networks.

Although only a few published studies have examined the COVID-19 pandemic's impact on the mental health of Filipino youth, existing research suggests a substantial burden. For instance, Montano and Abcedes (2020) surveyed 421 Filipinos aged 15–65 and found that 53.1% of respondents reported experiencing MSDS. Students and young adults – especially those who were unemployed – experienced the highest levels of COVID-19-related stress. More recently, Miranda and Tolentino (2023) found higher rates of depression among young Filipino women and participants who did not wish to disclose their gender. Although these studies provide valuable insights into the pandemic's impact on Filipino youth depression, neither was based on a nationally representative sample of young Filipinos.

In this study, we analyzed depression-specific data from two nationwide surveys on the health of Filipino youth, one conducted during the COVID-19 pandemic and another 8 years before the pandemic. Specifically, we addressed the following research questions: (1) What is the prevalence of MSDS among Filipino youth living in the midst of the COVID-19 pandemic and how does this compare to the MSDS prevalence 8 years before the pandemic? (2) What is the impact of the COVID-19 pandemic on MSDS prevalence across key sociodemographic characteristics?

## Methods

This study involves secondary data analysis and does not require institutional ethics review (Canadian Institutes of Health Research, 2018). Data access was facilitated by the Philippine Population Data Archive with permission from the University of the Philippines Population Institute and the Demographic Research and Development Foundation, Inc.

### Study population

Data examined in this study were from the Philippines – a democratic country in Southeast Asia with a presidential form of government and a population of about 109 million as of 2020. Up to 18.8% of the country's population are youth aged between 15 and 24 years, with a nearly even distribution of males (51.5%) and females (48.9%). The majority of the population reported Roman Catholicism (78.8%) as their religious affiliation, while the rest were affiliated with Islam (6.4%) or various other Christian denominations (Philippine Statistics Authority, 2024). The World Bank (2023) puts the Philippines in the lower- middle-income category, with a gross national income *per capita* of US$4,230.

We used data from the 2021 and 2013 survey rounds of the Young Adult Fertility and Sexuality Study (YAFS), a nationwide probability survey of Filipino youth aged 15–24 years. The 2021 YAFS survey (YAFS5) was conducted by the University of the Philippines Population Institute (UPPI) between August 2021 and January 2022, with funding from the Philippine Department of Health. The 2013 YAFS survey (YAFS4) was conducted by the UPPI and the Demographic Research and Development Foundation, Inc. (DRDF) between December 2012 and May 2013, with funding from the Australian government. The YAFS5 had a sample size of 10,949, while the YAFS4 had a sample size of 19,178. A description of the complex, multi-stage sampling strategy employed in these surveys is available elsewhere (DRDF and UPPI, 2016).

### Variables

Both the YAFS4 and YAFS5 surveys collected information on depressive symptoms using comparable versions of the Center for Epidemiological Studies – Depression (CES-D) Scale originally developed by Radloff (1977). Various versions of the CES-D exist and have been used in different studies. Kohout et al. (1993) tested two shortened CES-D versions and found that an 11-item version with a 3-response category, labeled as the Iowa version, had a substantially lower response burden while still capturing the same symptom dimensions with comparable reliability to Radloff's (1977) original 20-item scale. The YAFS5 adopted the Iowa version of the scale (Kohout et al., 1993) containing 11 items that used second-person lead-in pronouns (e.g., "Your appetite was poor"). Depression scale items in the YAFS4 started with "I" and "my" and included an item that was not in the YAFS5 ("I felt hopeful about the future"). The scale items in both surveys used the same three response categories (0 – rarely/not at all, 1 – sometimes, 2 – often). To enable direct comparison, only the 11 items present in both YAFS4 and YAFS5 were included in the analysis. A listing of the CES-D items used in YAFS4 and YAFS5 can be found in the online supplement.

The CES-D scale, along with the other questions in the YAFS questionnaires, were prepared in English and then translated

into the major Philippine languages including Tagalog, Cebuano, Ilonggo, Waray, Ilocano and Bicol. The translations were then translated back to English to ensure that the Philippine languages were faithful to the original English version (DRDF and UPPI, 2016).

Differences in depression scores were examined with respect to respondents' age, sex, education, marital status and socioeconomic status based on household wealth quintile, urbanity of place of residence, and sexual orientation, gender identity and expression (SOGIE). Categories of educational level were based on the Philippine Standard Classification of Education (Philippine Statistics Authority, 2017), while the YAFS5 research team calculated wealth index based on ownership of household amenities and vehicles and housing characteristics such as type of toilet facilities, following standard approaches developed by Rutstein and Johnson (2004) and Rutstein (2008).

There were no missing data for the CES-D scale and respondents' age, sex, education, marital status, place of residence and SOGIE and wealth index. Additionally, the survey had a considerable response rate (76% for the YAFS5), and nonresponse was accounted for in the calculation of sampling weights (Kabamalan and Marquez, in press).

## Analysis

Respondents with CES-D-11 sum-scores (minimum = 0; maximum = 22) greater than one standard deviation above the mean were identified as experiencing MSDS. This was based on a previous study (Puyat et al., 2021) that identified a pragmatic cutpoint for approximating the threshold (>20 sum-score) for clinical depression adopted by other researchers that used the original 22-item CES-D instrument (Kohout et al., 1993).

We plotted the prevalence estimates and the 95% confidence intervals to illustrate the change between 2013 and 2021 and highlight the magnitude of the differences between sociodemographic groups. In groups with more than two categories, that is, income, education and marital status, we only plotted the estimates from the lowest and highest categories (e.g., poorest vs. wealthiest income quintile).

We generated unadjusted log-binomial regression models to quantify the observed relative differences between groups. Prevalence ratios (PRs) from these models were interpreted as measures of disparity. Specifically, PR > 1 means that a group (e.g., females) had a higher MSDS prevalence compared to a reference group (e.g., males) and PR <1 means the opposite. To determine if the differences with respect to a characteristic (e.g., sex) are not due to differences in other characteristics (i.e., age, education, etc.), we ran multivariable log-binomial regression models adjusting for all the other sociodemographic characteristics.

Survey weights provided by the UPPI were applied throughout the analysis, which we performed in R version 4.4.1 (R Core Team, 2024) and R survey package 4.4-2 (Lumley et al., 2024).

## Results

### Sample and population characteristics

Table 1 describes the YAFS5 sample. There are slightly more males and youth aged 15–19, and about one in four youths had college education. In YAFS5, 56% of individuals aged 15–19 and 20.6% of those aged 20–24 reported being in school at the time of the survey. Similarly, in YAFS4, 53.7% of individuals aged 15–19

**Table 1.** Sample and population characteristics

| Characteristics | Sample (n) | Survey weighted distribution | | |
| --- | --- | --- | --- | --- |
| | | % | 95%CI | |
| **Sex** | | | | |
| Male | 5,312 | 48.8 | 47.6 | 50.0 |
| Female | 5,637 | 51.2 | 50.0 | 52.4 |
| **SOGIE** | | | | |
| Cisgender/heteronormative | 10,058 | 91.4 | 90.7 | 92.1 |
| Non-cisgender/homonormative | 885 | 8.6 | 7.8 | 0.1 |
| Not reported | 6 | <.05 | <.01 | 0.1 |
| **Age Group** | | | | |
| 15–19 | 6,490 | 58.9 | 57.8 | 60.1 |
| 20–24 | 4,459 | 41.1 | 39.9 | 42.2 |
| **Education** | | | | |
| No schooling/elementary | 750 | 7.0 | 6.3 | 7.7 |
| High school undergraduate | 4,408 | 40.2 | 38.9 | 41.5 |
| High school graduate/vocational | 3,132 | 28.5 | 27.4 | 29.6 |
| College or higher | 2,659 | 24.3 | 23.1 | 25.5 |
| **Marital status** | | | | |
| Never married | 9,288 | 84.7 | 83.6 | 85.9 |
| Living-in | 1,259 | 11.5 | 10.5 | 12.5 |
| Formally married | 345 | 3.3 | 2.8 | 3.8 |
| Separated/widowed/divorced | 57 | 0.5 | 0.3 | 0.7 |
| **Residence** | | | | |
| Rural | 8,252 | 66.9 | 66.5 | 67.3 |
| Urban | 2,697 | 33.1 | 32.7 | 33.5 |
| **Wealth index (quintile)** | | | | |
| Poorest | 1888 | 15.0 | 13.7 | 16.3 |
| Second | 2,326 | 18.1 | 16.9 | 19.3 |
| Middle | 2,330 | 18.9 | 17.7 | 20.2 |
| Fourth | 2,300 | 22.2 | 20.7 | 23.7 |
| Wealthiest | 2,105 | 25.7 | 24.0 | 27.5 |

Data source: 2021 Young Adult Fertility and Sexuality Study.
Wealth index is a measure of socioeconomic status.

and 11.0% of those aged 20–24 were enrolled in school (not shown in the table).

A small share of the sample (15%) had ever been married, and most were residing in rural areas (67%). The distribution of respondents by sociodemographic characteristics is similar in YAFS4 and YAFS5, except for a few differences: there were more youth with higher education, never-married and living in urban areas in YAFS5 than in YAFS4.

### Depression scores

The mean depression CES-D-11 scores in the YAFS5 and YAFS4 were 7.2 (SD = 3.8) and 7.3 (SD = 3.5), respectively. Cronbach's alpha for CES-D-11 was 0.77 in YAFS5 and 0.75 in YAFS4. Based

on the mean and standard deviation of the depression scores in both samples, CESD-11 > 11 was used as the cutpoint for MSDS.

### Prevalence and relative differences in the prevalence of MSDS in 2021

In 2021, 20.9% of Filipino youth experienced MSDS (Table 2), and the prevalence was higher among females (24.3%), non-cisgender/homonormative (32.3%) youth, those in the 15–19 age group (21.8%), youth with no schooling or had elementary education only (26.5%), youth from the poorest income quintile (25.1%), youth residing in rural areas (21.4%) and youth who

were separated/widowed/divorced (41.3%). Disparities in MSDS prevalence across various groups persisted even after accounting for differences in other sociodemographic characteristics using multivariable log-binomial regression analyses. In particular, females had 41% higher MSDS prevalence than males (aPR = 1.41) and non-cisgender/homonormative youth had a 60% higher MSDS prevalence than cisgender/homonormative individuals (aPR = 1.60). Youth who were separated, widowed or divorced had more than twice the prevalence of MSDS compared with youth who were never married (aPR = 2.02). There were significant socioeconomic disparities in MSDS as demonstrated by wealth quintile differences, and more evidently

**Table 2.** Prevalence of moderate to severe depressive symptoms by demographic characteristics (2021, YAFS5)

| Characteristics | Prevalence[a] | | | Prevalence ratio[b] | | | | | |
|---|---|---|---|---|---|---|---|---|---|
| | % | 95%CI | | Unadjusted | 95%CI | | Adjusted[c] | 95%CI | |
| **All** | 20.9 | 19.9 | 21.9 | – | – | – | – | – | – |
| **Sex** | | | | | | | | | |
| Male | 17.3 | 16.0 | 18.6 | Reference | | | Reference | | |
| Female | 24.3 | 22.8 | 25.7 | **1.40** | **1.27** | **1.55** | **1.41** | **1.28** | **1.56** |
| **SOGIE** | | | | | | | | | |
| Cisgender/heteronormative | 19.8 | 18.8 | 20.8 | Reference | Reference | | | | |
| Non-cisgender/homonormative | 32.3 | 28.4 | 36.1 | **1.63** | **1.42** | **1.86** | **1.60** | **1.40** | **1.84** |
| Not reported | 36.1 | 0.0 | 86.9 | – | – | – | – | – | – |
| **Age group** | | | | | | | | | |
| 15–19 | 21.8 | 20.5 | 23.1 | Reference | | | Reference | | |
| 20–24 | 19.6 | 18.2 | 21.0 | 0.90 | 0.82 | 0.99 | 0.97 | 0.86 | 1.10 |
| **Education** | | | | | | | | | |
| No schooling/elementary | 26.5 | 22.6 | 30.4 | **1.59** | **1.33** | **1.90** | **1.56** | **1.28** | **1.90** |
| High school undergraduate | 22.1 | 20.6 | 23.6 | **1.33** | **1.17** | **1.51** | **1.29** | **1.10** | **1.51** |
| High school graduate/vocational | 21.4 | 19.5 | 23.2 | **1.28** | **1.12** | **1.46** | **1.26** | **1.08** | **1.47** |
| College or higher | 16.7 | 14.9 | 18.4 | Reference | | | Reference | | |
| **Marital status** | | | | | | | | | |
| Never married | 20.4 | 19.3 | 21.5 | Reference | | | Reference | | |
| Living-in | 22.1 | 19.3 | 25.0 | 1.08 | 0.94 | 1.25 | 0.99 | 0.84 | 1.16 |
| Formally married | 25.7 | 19.8 | 31.6 | 1.26 | 1.00 | 1.59 | 1.11 | 0.86 | 1.43 |
| Separated/widowed/divorced | 41.3 | 26.1 | 56.6 | **2.02** | **1.40** | **2.94** | **2.02** | **1.38** | **2.96** |
| **Residence** | | | | | | | | | |
| Rural | 21.4 | 20.2 | 22.6 | Reference | | | Reference | | |
| Urban | 19.9 | 18.1 | 21.7 | 1.08 | 0.97 | 1.19 | 1.01 | 0.90 | 1.12 |
| **Wealth index (quintile)** | | | | | | | | | |
| Poorest | 25.1 | 22.6 | 27.6 | **1.30** | **1.11** | **1.53** | **1.19** | **1.01** | **1.42** |
| Second | 21.8 | 19.8 | 23.9 | 1.13 | 0.97 | 1.32 | 1.09 | 0.92 | 1.29 |
| Middle | 22.6 | 20.3 | 24.8 | 1.17 | 1.00 | 1.37 | 1.15 | 0.97 | 1.35 |
| Fourth | 17.8 | 15.8 | 19.7 | 0.92 | 0.78 | 1.09 | 0.90 | 0.76 | 1.07 |
| Wealthiest | 19.3 | 16.9 | 21.6 | Reference | | | Reference | | |

[a]Prevalence (%) counts everyone with CESD-11 score > 11. Prevalence estimates and their 95% confidence intervals were generated using YAFS5 survey weights and parameters.
[b]Prevalence ratio is a relative measure of association derived from dividing the prevalence estimate of one group to a reference group.
[c]Adjusted prevalence ratio was estimated *via* log binomial regression, accounting for all the other listed covariates and survey weights and parameters. Wealth index is a measure of socioeconomic status.
Bold entries are statistically significant at p<0.05.

through a dose–response association between education and MSDS where youth with no schooling/elementary education had the highest prevalence of MSDS (aPR = 1.56) compared to youth with college education (Table 2).

### Changes in the prevalence of MSDS and disparities in MSDS

The prevalence and relative differences in the prevalence of MSDS in 2013 were reported and discussed in detail in a previous publication (Puyat et al., 2021). The prevalence estimates and results of the analyses of relative differences using the cutpoint obtained in this study are provided in Table 3.

As Tables 2 and 3 indicate, the prevalence of MSDS among Filipino youth surged from 9.6% in 2013 to 20.9% in 2021. This significant, twofold increase was observed across all sociodemographic groups (Figure 1), although certain groups experienced more pronounced rises. In particular, the prevalence of MSDS prevalence significantly increased among females (from 10.8% to 24.3%), non-cisgender/homonormative individuals (from 7% to 32.3%), those with no schooling or only elementary education (from 10.8% to 26.5%), youth from the poorest income quintile

**Table 3.** Prevalence of moderate to severe depressive symptoms by demographic characteristics (2013, YAFS4)

| Characteristics | Prevalence[a] | | | Prevalence ratio[b] | | | | | |
| --- | --- | --- | --- | --- | --- | --- | --- | --- | --- |
| | % | 95%CI | | Unadjusted | 95%CI | | Adjusted[c] | 95%CI | |
| **All** | 9.6 | 8.9 | 10.3 | – | – | – | – | – | – |
| **Sex** | | | | | | | | | |
| Male | 8.3 | 7.4 | 9.2 | Reference | | | Reference | | |
| Female | 10.8 | 9.8 | 11.9 | **1.31** | **1.13** | **1.51** | **1.36** | **1.17** | **1.58** |
| **SOGIE** | | | | | | | | | |
| Cisgender/heteronormative | 9.7 | 9.0 | 10.4 | Reference | Reference | | | | |
| Non-cisgender/homonormative | 7.0 | 4.8 | 9.3 | 0.72 | 0.52 | 1.00 | 0.72 | 0.52 | 1.00 |
| Not reported | 1.3 | 0.0 | 4.0 | | | | | | |
| **Age group** | | | | | | | | | |
| 15–19 | 9.7 | 8.9 | 10.5 | Reference | | | Reference | | |
| 20–24 | 9.5 | 8.4 | 10.6 | 0.98 | 0.85 | 1.12 | 1.08 | 0.92 | 1.26 |
| **Education** | | | | | | | | | |
| No schooling/elementary | 10.8 | 9.2 | 12.4 | **1.28** | **1.02** | **1.62** | **1.41** | **1.12** | **1.78** |
| High school undergraduate | 10.4 | 9.4 | 11.4 | **1.23** | **1.00** | **1.52** | **1.32** | **1.09** | **1.59** |
| High school graduate/vocational | 8.9 | 7.6 | 10.1 | 1.05 | 0.84 | 1.32 | 1.11 | 0.89 | 1.37 |
| College or higher | 8.4 | 6.9 | 9.9 | Reference | | | Reference | | |
| **Marital status** | | | | | | | | | |
| Never married | 9.5 | 8.8 | 10.3 | Reference | | | Reference | | |
| Living-in | 8.7 | 6.7 | 10.8 | 0.91 | 0.71 | 1.17 | 0.80 | 0.63 | 1.03 |
| Formally married | 9.8 | 8.1 | 11.4 | 1.02 | 0.85 | 1.23 | 0.89 | 0.73 | 1.08 |
| Separated/widowed/divorced | 18.3 | 11.6 | 25.1 | **1.92** | **1.31** | **2.81** | **1.60** | **1.07** | **2.40** |
| **Residence** | | | | | | | | | |
| Rural | 9.3 | 8.5 | 10.1 | Reference | | | Reference | | |
| Urban | 10.4 | 8.9 | 11.9 | 0.89 | 0.76 | 1.06 | 0.85 | 0.71 | 1.02 |
| **Wealth index (quintile)** | | | | | | | | | |
| Poorest | 10.6 | 9.4 | 11.8 | 1.02 | 0.81 | 1.29 | 0.99 | 0.79 | 1.25 |
| Second | 9.7 | 8.4 | 11.0 | 0.93 | 0.74 | 1.18 | 0.93 | 0.74 | 1.15 |
| Middle | 9.2 | 7.9 | 10.5 | 0.89 | 0.69 | 1.14 | 0.89 | 0.70 | 1.14 |
| Fourth | 8.3 | 6.9 | 9.6 | 0.79 | 0.62 | 1.02 | 0.79 | 0.62 | 1.02 |
| Wealthiest | 10.4 | 8.3 | 12.5 | Reference | | | Reference | | |

[a]Prevalence (%) of moderate/severe depressive symptoms counts everyone with CESD-11 score > 11. Prevalence estimates and their 95% confidence intervals were generated using YAFS4 (2013) survey weights and parameters.
[b]Prevalence ratio is a relative measure of association derived from dividing the prevalence estimate of one group by the prevalence estimate of a reference group.
[c]Adjusted prevalence ratios were estimated *via* log binomial regression, accounting for all the other listed covariates, survey weights and parameters. Wealth index is a measure of socioeconomic status.
Bold entries are statistically significant at p<0.05.

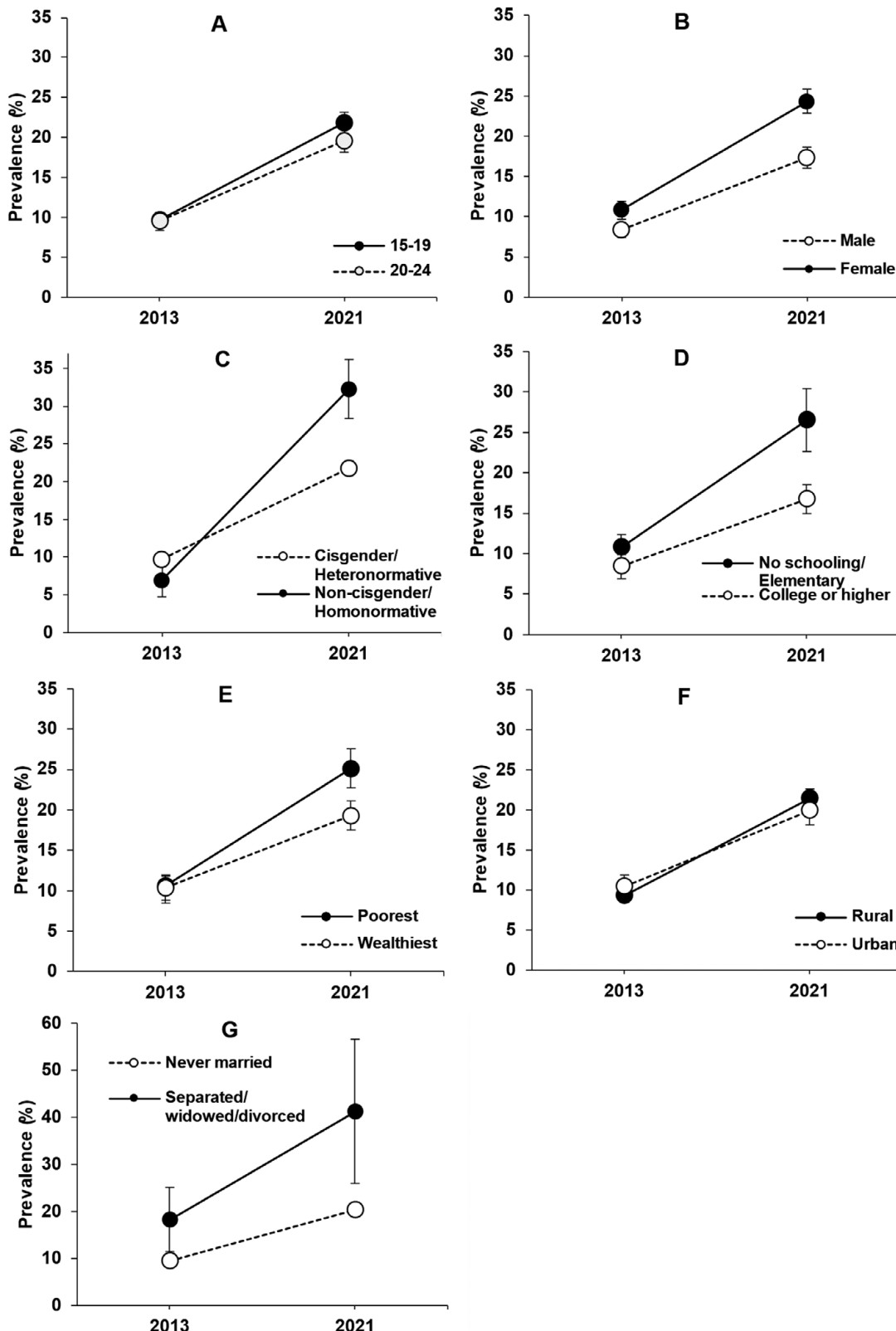

**Figure 1.** Prevalence (%) of moderate to severe depressive symptoms in 2013 and 2021 by (A) age; (B) sex; (C) sexual orientation, gender identity and gender expression; (D) highest education level, (E) wealth index, (F) urbanity, and (G) marital status.
Prevalence (%) estimates and their 95%CIs were generated using survey weights provided by UPPI; 95%CIs for all prevalence estimates were plotted even for those with intervals that are too narrow to show up in the plots. In groups with multiple categories, only the highest and lowest categories were shown.
Wealth index is a measure of socioeconomic status.

(from 10.6% to 25.1%) and those who were separated, widowed or divorced (from 18.3% to 41.3%).

The comparison of adjusted PRs confirmed that the increase in MSDS disparities in 2021 was independent of variations in other sociodemographic characteristics. Specifically, sex disparities widened from aPR = 1.36 to aPR = 1.40, indicating that females experienced a greater increase in MSDS prevalence than males in 2021 (Tables 2 and 3). Gender disparities also widened, rising from aPR = 0.72 in 2013 to aPR = 1.60 in 2021, suggesting a substantially higher increase in MSDS among non-cisgender/homonormative compared to cisgender/heteronormative youth. Additionally, disparities by education grew from aPR = 1.41 to aPR = 1.59, highlighting a substantial rise in MSDS among youth with less than secondary education in 2021. Notably, while wealth index disparities were not significant (aPR = 0.99) in 2013, the 2021 results reveal that youth in the poorest wealth quintile (aPR = 1.19) experienced the largest increase in MSDS prevalence compared with those in the richest households (Table 3).

## Discussion

Consistent with studies that documented a doubling of the prevalence of depression among youth during the COVID-19 pandemic (Racine et al., 2021), our analysis of data from two nationwide probability surveys of Filipino youth aged 15–24 years indicate that the prevalence of MSDS rose from about 10% in 2013 to about 21% in 2021. Our findings also indicate that sociodemographic disparities in MSDS have worsened in 2021, suggesting a greater burden of MSDS among specific groups, including females, non-cisgender or homonormative individuals, youth with no schooling or only elementary education, youth from the poorest income quintile and individuals who were separated, widowed or divorced.

This sudden and significant increase in the prevalence of and disparities in MSDS could largely be attributed to the nature of the COVID-19 pandemic as a global disaster. Disasters exacerbate preexisting inequalities and inefficiencies while creating new vulnerabilities. The pandemic exposed people and communities to higher levels of stress and crisis while greatly reducing access to support and services (Gaiser et al., 2023). This is especially true for marginalized groups that are more likely to experience social inequalities.

In the Philippines, the government's response to the pandemic consisted of near-total lockdowns – among the most stringent in the world (Herreros and Svendsen, 2024) – which affected livelihood, especially those of lower socioeconomic status and restricted access to all levels of health care and to sources of socio-emotional support. Lockdown measures did not take into account income, livelihood, food security, space and population density (Hapal, 2021), thus, affecting mostly the marginalized groups and those with fewer resources and less access to mental health support. For students, these lockdowns led to prolonged school closures, which were associated with significant learning loss (Patrinos et al., 2023). As schools eventually reopened, classes shifted to remote learning, which required access to the internet and appropriate devices. This not only increased school expenses but also deprived students of the structure and support that in-person classes typically provide. Being indoors also decreased opportunities for social interaction and physical exercise, including sports and play, which are significant promotive factors for the youth's wellbeing (Andersen et al., 2021; Kemel et al., 2022; Takiguchi et al., 2022).

Although COVID-19 contributed to the significant increase in the prevalence and sociodemographic disparities in MSDS, it is also likely that depression among Filipino youth was already on the rise before the pandemic. This aligns with studies from other countries, including the United States, which have reported upward trends in youth depression over the past decade (Daly, 2022; Keyes et al., 2024). Various explanations have been proposed for this trend, such as the widespread use of digital technology and social media, but findings from meta-analysis and systematic reviews suggest that these exposures have only small to negligible effects on youth depression (Appel et al., 2020; Ferguson et al., 2022; Hancock et al., 2022; Valkenburg et al., 2022). Another potential driver for the rising trend is the growing number of 'left-behind' children (those whose parents migrated), who are at higher risk of depression (Fellmeth et al., 2018).

We do not know if the MSDS experienced by respondents in the two surveys were self-limiting. A meta-analysis of antidepressant clinical trials involving children and adolescents (Meister et al., 2020) found a pooled estimate of clinician-rated placebo rates of about 45% (95%CI: 41–50%), indicating that a portion of youth with MSDS were either self-limiting or may recover without pharmacological intervention. Other studies, on the other hand, underscore the persistence of MSDS. One recent longitudinal analysis (Keyes et al., 2024), for example, found that among youth borne between 1997 and 2001 who exhibited high depressive symptoms at age 18, about 45.6% and 46.3% continued to report high depressive symptoms at ages 19–20 and 21–22 years, respectively. These findings suggest that a substantial proportion of youth with MSDS – roughly 5–6 out of 10 – could benefit from early intervention and prevention through various forms of mental health services, treatments and support. This is particularly crucial given evidence linking adolescent depressive symptoms with a higher risk of adult depression (Gustavson et al., 2018) and impaired functioning (Weersing et al., 2017).

The study's findings on the disproportionate burden of depression among marginalized groups highlight the critical need to expand the availability and accessibility of a broad range of mental health services, as outlined in the Philippines' 2018 Mental Health Act. A pre-pandemic assessment of the state of mental health care in the Philippines revealed a severe shortage of mental health professionals in the country, particularly in government-funded facilities, with the majority of services being delivered in hospital settings (Lally et al., 2019). That report also underscored the underdevelopment of community-based mental health services, thereby leaving large gaps in accessible care (Lally et al., 2019).

Incidentally, the onset of the COVID-19 pandemic may have accelerated the adoption of teletherapy as a practical and scalable method of delivering mental health services. Teletherapy has long been shown to be as effective as in-person therapy (Lin et al., 2022) and is beneficial for treating depression and other mental disorders (Varker et al., 2019). Specific guidelines are, therefore, needed to ensure teletherapy's effectiveness (Sablone et al., 2024) and to prevent further exacerbation of existing mental health treatment disparities. In the Philippines, access to teletherapy is largely limited to those with sufficient financial, technological resources and private spaces, making it inaccessible for many low-income households and out-of-school youth. To address this gap, there is an urgent need to develop alternative mental health services, particularly those that are based in the community and led or supported by peers.

Our analysis focused on the SDH and did not include mediating variables or psychological constructs, such as resilience. Given

this scope, we aimed to avoid extrapolating findings beyond the variables we analyzed. However, we acknowledge that constructs, such as resilience, provide an important lens through which to interpret the experiences of Filipinos during the pandemic.

Some studies have highlighted that Filipino resilience is deeply rooted in a strengthened sense of *kapwa* or connectedness with others (Macaraeg and Bersamira, 2024; Pacaol and Siguan, 2024). These studies suggest that Filipinos enhance their resilience by nurturing relationships with friends, family and their broader community. We note that pandemic-related disruptions, such as school closures, mobility restrictions and the quarantine of family members, significantly constrained opportunities for such relationship-building. These disruptions may have, in turn, influenced the ability of individuals and communities to mobilize resilience during this period.

Future research should examine the role of resilience and related constructs as mediating factors in the experience of depression and access to mental health services during pandemics. Investigating how resilience and similar constructs mediate the effects of social and structural barriers on mental health outcomes could provide valuable insights for designing interventions that strengthen community-based support systems and improve mental health service delivery in times of crisis.

This study offers valuable insights into the magnitude of the burden of depression among Filipino youth and the characteristics of the groups that are disproportionally affected. The study's estimate provides essential data to inform policy, clinical practice and future mental health service planning. Nevertheless, several limitations need to be acknowledged. First, our findings were based on two cross-sectional surveys, which limited our ability to infer about the causal impact of COVID-19 on depression prevalence. Second, the reliance on self-reported data to assess MSDS may have introduced recall and social desirability biases, potentially impacting the accuracy of our estimates. Additionally, slight differences in the CES-D scale items between the two survey cycles may have influenced the respondents' answers. Third, as the YAFS5 data were collected during the height of the COVID-19 pandemic, we cannot fully discount the potential effects that public health measures and survey restrictions may have had on responses. Fourth, although we accounted for various variables in our regression models, residual confounding due to unmeasured variable factors may still have affected our findings. Finally, while our use of the mean and standard deviation to derive an MSDS threshold yielded near identical cutpoints for 2013 and 2021, a validated version of the CES-D scale and MSDS cutpoints specific to Filipino youth would have been preferable.

## Conclusion

The prevalence of MSDS among Filipino youth rose from about 10% in 2013 to 21% in 2021, disproportionately impacting females, non-cisgender or homonormative individuals, those who were separated, divorced or widowed, those with no schooling or with elementary education only and youth from the poorest income groups. Expanding access to quality mental health services is needed to help alleviate depressive symptoms and to mitigate its long-term effects, particularly among the underserved and marginalized groups. Follow-up studies, including longitudinal surveys, are needed to assess whether the increase in prevalence between 2013 and 2021 will persist in the coming years.

Additionally, future research should explore the types of support and services received by Filipino youth who experienced MSDS during the pandemic and evaluate their long-term outcomes.

**Open peer review.** To view the open peer review materials for this article, please visit http://doi.org/10.1017/gmh.2025.39.

**Supplementary material.** The supplementary material for this article can be found at http://doi.org/10.1017/gmh.2025.39.

**Data availability statement.** The data analyzed in this study are available for access from the University of the Philippines Population Institute (UPPI), the Philippine Population Data Archive and the Demographic Research and Development Foundation, Inc. Researchers interested in obtaining the data can communicate with these organizations to request access and review any applicable usage guidelines.

**Acknowledgments.** The authors would like to thank the University of the Philippines Population Institute (UPPI), Philippine Population Data Archive and the Demographic Research and Development Foundation, Inc. for facilitating and granting access to the data for this study. The views, findings, opinions and conclusions expressed herein are solely the authors' and do not necessarily represent the views of any of these organizations.

**Author contribution.** Conceptualization: JHP, DLS, ACT, SDA; Formal analysis: JHP; Methodology: JHP, DLS, ACT, SDA; Visualization: JHP; Writing-original draft: JHP, DLS, ACT; Writing-review and editing: JHP, DLS, ACT, SDA. All authors have read and approved the final version of the manuscript.

**Financial support.** JHP was supported by the Michael Smith Health Research British Columbia Scholar Awards (#18299).

**Competing interests.** The authors declare no potential conflict of interest with respect to the research, authorship and publication of this manuscript.

**Ethical statement.** This study utilizes secondary data from two nationwide health surveys conducted in 2013 and 2021. According to the 2018 Canadian Tri-Council Policy Statement (TCPS), formal ethical review is not required for secondary data analysis when data are de-identified and used for research purposes. The original data collection adhered to ethical standards, ensuring informed consent and participant confidentiality.

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
