## [Reviewer Report]

The authors of this manuscript provided an interesting topic - how youth depressive symptoms changed before and during the COVID-19 pandemic, and the authors provided some insights into their investigation using two secondary data analyses. There are some areas for improvement for this manuscript to be addressed to be published. I’d also encourage authors to go through their manuscripts again to ensure coherence and cohesion. My comments are listed below.

1. In the introduction section, the authors could have expanded on the aspects of how the pandemic has impacted youths‘ mental health, for instance, quarantine, lack of social communication support from peers, etc and how that linked to the deteriorated youths’ MSDS in the Philippines.

2. PG11, lines 32-33, an acronym “SES” was used. It would have been better to give a full explanation (e.g., socioeconomic status, SES) and an abbreviation, as this acronym has not been provided elsewhere throughout the manuscript.

3. The authors could have discussed more about the instruments used for data collection. Have the YAFS4 and YAFS5 been adopted for the use targeting the Filipino population, or did they use the original scale (i.e., the English version)? Having checked the original CES-D (20-item), the scale used a 4-point scale ranging from 0-4, which differed from what the authors discussed, a 3-point scale, ranging from 0 to 2. What caused this distinction?

Reference to the scale: https://www.apa.org/depression-guideline/epidemiologic-studies-scale.pdf

4. In the discussion section, the authors discussed the possibility of how the pandemic has impacted students, which may have led to poor or deteriorated mental health. However, there was limited information in the introduction and methodology (particularly the sample and/or population included in the present study) regarding the size of the student population included, thus, whether student-specific factors contributed to deteriorated depressive symptoms due to the pandemic remains unclear. The authors could have expanded on this part in the revision.

5. Table 2 and Table 3 both showed that the youths with no or little education and those who were poorest at the wealth index had the highest prevalence of depressive symptoms. How did the research team that collected this data ensure the accuracy of both groups? How did the authors of this paper (or those who cleaned the data) deal with missing data? There is a lack of information regarding the quality of this data.

---

## [Reviewer Report]

Review of the Article: “Rising Prevalence of Depression and Widening Sociodemographic Disparities in Depressive Symptoms among Filipino Youth: Findings from Two Large Nationwide Cross-Sectional Surveys”

1. Terminology Adjustment - The authors might consider replacing the term “low-resource settings like the Philippines” with a more accurate descriptor, as the Philippines is rich in natural resources. This could help clarify the context of the discussion around mental health resources.

2. Theoretical Framework - It would enhance the study to incorporate a theoretical framework that is relevant to the Filipino context. This could provide a deeper understanding of the underlying factors affecting depressive symptoms among Filipino youth.

3. Comparison with Existing Literature - I recommend that the authors compare their findings with other studies focused on Filipino resilience. Discussing these aspects would provide valuable insights and could enrich the discussion of the results, especially given the scope of the two large nationwide surveys conducted.

4. Understanding Filipino Psyche - Lastly, it would be beneficial for the authors to explore how their findings contribute to our understanding of the Filipino psyche and related mental health concepts within the Philippine context. This additional discussion could offer a more nuanced perspective on the implications of the study’s results.

---

## [Editor Report]

Dear Authors,

Your manuscript: “Rising Prevalence of Depression and Widening Sociodemographic Disparities in Depressive Symptoms among Filipino Youth: Findings from Two Large Nationwide Cross-Sectional Surveys,” has now been reviewed.

---

## [Editor Report]

Dear Joseph Puyat,

Your revised manuscript “Rising Prevalence of Depression and Widening Sociodemographic Disparities in Depressive Symptoms among Filipino Youth: Findings from Two Large Nationwide Cross-Sectional Surveys” has now been reviewed,